# Neurosymbolic Deep Generative Models for Sequence Data with Relational Constraints

## Abstract

Recently, there has been significant progress designing deep generative models that generate realistic sequence data such as text or music. Nevertheless, it remains difficult to incorporate high-level structure to guide the generative process. We propose a novel approach for incorporating structure in the form of relational constraints between different subcomponents of an example (e.g., lines of a poem or measures of music). Our generative model has two parts: (i) one model to generate a realistic set of relational constraints, and (ii) a second model to generate realistic data satisfying these constraints. To train model (i), we propose a novel program synthesis algorithm that infers the relational constraints present in the training data, and then train the models based on the resulting relational constraints. In our experiments, we show that our approach significantly improves over state-of-the-art approaches in terms of capturing high-level structure in the data, while performing comparably or better in terms of low-level structure.

## 1 Introduction

Over the past few years, there has been tremendous progress in designing deep generative models for generating sequence data such as natural language (Vaswani et al., 2017) or music (Huang et al., 2019). These approaches leverage the vast quantities of data available in conjunction with unsupervised and self-supervised learning to learn probabilistic models of the data; then, new examples can be generated by sampling from these models, with the possibility of conditioning on initial elements of the sequence.

Despite this progress, a key challenge facing deep generative models is the difficulty incorporating high-level structure into the generated examples—e.g., rhyming and meter across lines of a poem, or repetition across measures of a piece of music. The ability to capture high-level structure is important for improving the quality of the generated data, especially in low-data regimes where only small numbers of examples are available—intuitively, knowledge of the structure compresses the amount of information that the generative model has to learn. Furthermore, *explicit* representations of structure—i.e., in a symbolic way rather than implicitly in a vector embedding—can have the added benefit that users can modify the structure to guide the generative process.

Recently, Young et al. (2019) proposed a technique called *neurosymbolic generative models* for incorporating high-level structure into image generation, focusing on simple 2D repeating patterns in images of building facades (e.g., repeating windows). The basic idea is to leverage *program synthesis* to extract structure from data—in particular, given an example image $x$, they devise an algorithm $\mathcal{A}$ that extracts a program $c = \mathcal{A}(x)$ that represents the set of 2D repeating patterns present in training examples $x$. Then, using the pairs $(x, c)$, they train two generative models: (i) a model $p_\phi(c)$ that generates a program, and (ii) a model $p_\theta(x \mid c)$ that generates an image that contains the structure represented by $c$.

However, their approach is heavily tailored to the image domain in several ways. First, their representation of structure is geared towards relatively simple patterns occurring in images of building facades. In addition, their algorithm $\mathcal{A}$ is specifically designed to extract this kind of program from an input image, as are their models $p_\phi(c)$ for generating programs and $p_\theta(x \mid c)$ for generating images conditioned on the program.

We represent relational constraints $c_x$ present in an example $x$ by relating each subcomponent $w$ of a given example $x$ with a *prototype* $\tilde{w}$, which can intuitively be thought of as the "original" subcomponent from which $w$ is constructed. In particular, the relationship between $w$ and $\tilde{w}$ is labeled with a set of relations $R$, which encodes the constraint that $w$ and $\tilde{w}$ should satisfy relation $r$ for each $r \in R$. Importantly, while each subcomponent is associated with a single prototype, each prototype may be associated with multiple subcomponents. As a consequence, different subcomponents associated with the same prototype are related in some way. This representation is compact, only requiring linearly many constraints in the number of subcomponents in $x$ (assuming the number of prototypes is constant). Intuitively, compactness ensures the representation both generalizes well and is easy to generate.

Then, we design a program synthesis algorithm that can extract an optimal representation of the structure present in a training example $x$. We show how to express the synthesis problem as a constrained combinatorial optimization problem, which we solve using an SMT solver Z3 (De Moura & Bjørner, 2008). Next, we represent $c$ as a sequence, and design $p_\phi(c)$ to be inferred through a specialized sequence VAE. Finally, we propose three possible designs of $p_\theta(x \mid c)$ based on trying to identify an example $x$ that is realistic (e.g., according to a pretrained generative model $p_\theta(x)$) while simultaneously satisfies the constraints $c$.

We evaluate our approach on two tasks: poetry generation, where the relational constraints include rhyming lines or lines with shared meter, and music generation, where the relational constraints include equality in terms of pitch or rhythm, that one measure is a transposition of another (i.e., pitches shifted up or down by a constant amount), etc. We show that our approaches outperform or perform similarly to SOTA models according to many metrics.

Finally, we also perform a qualitative evaluation where we show how the user can modify the high-level to generate examples that satisfy additional desired constraints. This ability demonstrates an important feature of our approach—i.e., that the user can guide the generative process by modifying the relational constraints as desired.

**Example.** Figure 1 illustrates how our approach is applied to generate poetry. During training, our approach uses program synthesis to infer relational constraints $c_x$ present in the examples $x$, and uses both $x$ and $c_x$ to train the generative models. Here, $c_x$ is a bipartite graph, where the LHS vertices are *prototypes*, and the RHS vertices correspond to lines of $x$. Each vertex on the right is connected to exactly one prototype, and is labeled with constraints on how it should relate to its prototype. To generate new examples, it first samples relational constraints $c$, and then samples an example $x$ that satisfies $c$—i.e., we need to choose a line to fill each RHS node in a way that the line satisfies the relations with its prototype. Furthermore, a user can modify the sampled constraint $c$ to guide the generative process. Thus, our approach enables users to flexibly incorporate domain knowledge on the high-level structure of the data into the generative process, both in terms of the relational constraints included and by allowing them to modify the generated relational constraints.

**Related work.** There has been recent interest in leveraging program synthesis to improve machine learning. For instance, it has been applied to unsupervised learning of latent structure in drawings (Ellis et al., 2015) and to reinforcement learning (Verma et al., 2018). These techniques have benefits such as improving interpretability Verma et al. (2018); Ellis et al. (2020), enabling learning from fewer examples (Ellis et al., 2015), generalizing more robustly (Inala et al., 2019), and being easier to formally reason about (Bastani et al., 2018). More recently, there has been work leveraging program synthesis in conjunction with deep learning, where the DNN handles perception and program synthesis handles high-level structure (Ellis et al., 2017), including work in the lifelong learning setting (Valkov et al., 2018). In contrast to these approaches, our focus is on generative models. In particular, we extend recent work leveraging these ideas in the setting of image generation to incorporating high-level relational structure into sequence generation tasks (Young et al., 2019).

Early music generation approaches were rule-based (Ovans & Davison, 1992) or used simple statistical models such as Markov models (Sandred et al., 2009; Cope, 1987) or probabilistic CFGs (Quick, 2016). Recent work has used deep learning to generate music (Huang et al., 2019; OpenAI, 2019) and poetry (Liao et al., 2019); our experiments show that these approaches have difficulty generating realistic high-level structure. Approaches have incorpo-

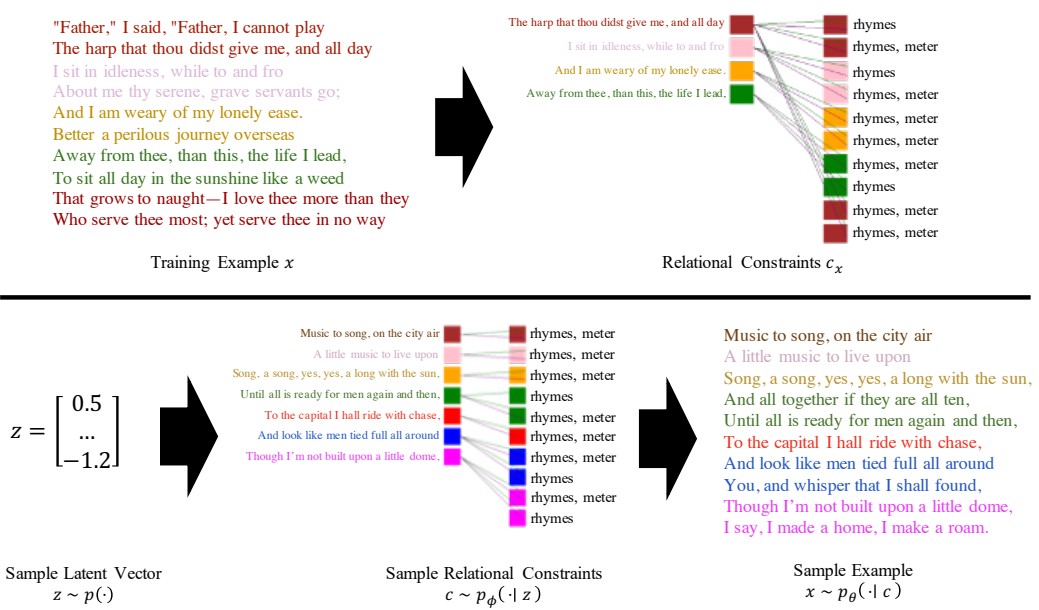

Figure 1: Top: Process for training. For each training example $x$, our algorithm uses program synthesis to infer the relational constraints $c_x = \mathcal{A}(x)$ present in $x$. Then, it (i) uses $c_x$ to train $p_\phi(c) = \mathbb{E}_{z \sim p(\cdot)}[p_\phi(c \mid z) \cdot p(z)]$, and (ii) uses $(c_x, x)$ to train $p_\theta(x \mid c)$. Bottom: Process for generating a sample $x$ from the learned models $p_\phi(c \mid z)$ and $p_\theta(x \mid c)$. Lines with the same prototype are shown in the same color; metrical constraints are represented as purple and rhyme constraints as green edges.

rated structure into deep learning to generate music (Medeot et al., 2018) or poetry (Castro & Attarian, 2018), but they are domain specific; we find they do not perform at a human level on capturing global (and sometimes local) structure. Some approaches incorporate expert-provided constraints such as rhyme and meter to generate poetry (Lau et al., 2018); unlike our approach, they cannot automatically learn and generate these constraints from data.

## 2 BACKGROUND ON NEUROSYMBOLIC GENERATIVE MODELS

Consider the problem of learning a generative model given training data from the underlying distribution. Given training examples $x_1, ..., x_k \sim p^*$, our goal is to learn a generative model $p_\theta \approx p^*$ from which we can draw additional samples $x \sim p_\theta$. We consider sequence data—i.e., an example $x \in \mathcal{X}$ is a sequence $x = (w_1, ..., w_m) \in \mathcal{W}^m$.[1] For example, each *subcomponent* $w$ may be a line of a poem or a measure of music, and $x$ may be a poem or song.

We are interested in domains where likely examples satisfy latent relational constraints $c \in \mathcal{C}$ over the subcomponents. For instance, $c$ may say that two measures $w_i$ and $w_j$ of $x$ start with the same series of pitches, or two lines $w_i$ and $w_j$ of $x$ rhyme. We assume given a set of relations $\mathcal{R}$ (e.g., $r \in \mathcal{R}$ might be "rhyme" or "equal"), and a function $f : \mathcal{W} \times \mathcal{W} \times \mathcal{R} \to \mathbb{B}$ (where $\mathbb{B} = \{0, 1\}$) such that $f(w, w', r)$ indicates whether $w$ and $w'$ satisfy relation $r$. Then, $c$ is a compact representation of the relations present in an input $x$. We describe the structure of $c$ in detail in Section 3.1; for now, the approach we describe works for any choice of $c$. In particular, we build on *neurosymbolic generative models* (Young et al., 2019),

---

[1]We use a fixed $m$ to simplify our exposition; our approach trivially extends to variable $m$.

where $c$ is itself generated based on a latent value $z \in \mathcal{Z}$—i.e.,

$$p_{\theta,\phi}(x) = \int \sum_{c \in \mathcal{C}} p_\theta(x \mid c) \cdot p_\phi(c \mid z) \cdot p(z) dz.$$

Then, Young et al. (2019) considers the variational distribution

$$q_{\tilde{\phi}}(c, z \mid x) = q_{\tilde{\phi}}(z \mid c) \cdot q(c \mid x) \qquad \text{where} \qquad q(c \mid x) = \delta(c - c_x).$$

Here, $\delta$ is the Dirac delta function, and $c_x$ is a single representative associated with $x$. In particular, $c_x$ is generated from $x$ using a program synthesis algorithm David & Kroening (2017)—i.e., an algorithm $\mathcal{A}$ that takes as input an example $x$ and outputs a program $c = \mathcal{A}(x)$ encoding the relational constraints present in $x$. Next, Young et al. (2019) derive an evidence lower bound

$$\log p_{\theta,\phi}(x) \geq \log p_\theta(x \mid c_x) + \mathbb{E}_{q_{\tilde{\phi}}(z \mid c_x)}[\log p_\phi(c_x \mid z)] - D_{\mathrm{KL}}(q_{\tilde{\phi}}(z \mid c_x) \parallel p(z)). \qquad (1)$$

where $D_{\mathrm{KL}}$ is the KL divergence and $H$ is the information entropy. The first term of (1) is the log-likelihood of a generative model predicting the probability of example $x$ given relational structure $c_x$, and the second and third terms form the loss of a variational autoencoder (VAE) $p_\phi(c \mid z)$ and $q_{\tilde{\phi}}(z \mid c)$ (Kingma & Welling, 2019). In summary, this approach separately learns (i) a VAE to generate $c$ given $z$, and (ii) a generative model to generate $x$ given $c$; the latter can be a second VAE or a generative adversarial network (GAN) (Goodfellow et al., 2014). This approach is called *synthesis-guided generative models (SGM)* since it uses program synthesis to guide training.

To leverage this framework, we have to instantiate (i) the space of relational constraints $\mathcal{C}$, (ii) the synthesis algorithm $\mathcal{A} : \mathcal{X} \to \mathcal{C}$ used to extract a program encoding the structure of $x$, and (iii) the architectures of $p_\phi(c \mid z)$, $q_{\tilde{\phi}}(z \mid c)$, and $p_\theta(x \mid c)$. In previous work, Young et al. (2019) used heuristics specific to the the image domain to achieve these goals—in particular, they used (i) simple equality constraints on sub-regions of the image designed to capture 2D repeating patterns, (ii) a custom synthesis algorithm that greedily adds constraints in the data to the program, and (iii) a representation of $c_x$ as an image, in which case $p_\theta$ is a generative model over images, and $p_\phi, q_{\tilde{\phi}}$ based on an encoding of $c$ as a fixed-length vector.

We design a synthesis algorithm that expresses the synthesis problem as a constrained combinatorial optimization problem, which it solves using Z3 (De Moura & Bjørner, 2008). In terms of (iii), our programs encode declarative constraints rather than imperative renderings, so the previous architectures of $p_\phi$, and $q_{\tilde{\phi}}$ cannot be used. Instead, we use expert domain-specific heuristics, transformers (Vaswani et al., 2017), or graph neural networks (GNNs) (Kipf & Welling, 2017) for $p_\phi$ and $q_{\tilde{\phi}}$. For $p_\theta$, we propose several methods for imposing the constraints encoded by $c$ when generating an example $x$.

## 3   Relational Constraints for Sequence Data

In this section, we describe how we represent relational constraints $r$, as well as our algorithm $\mathcal{A}$ for synthesizing the relational constraints $c_x = \mathcal{A}(x)$ present in an example sequence $x$.

### 3.1   Graph Representation of Relational Constraints

Recall that our generative model operates by first generating a relational program $c$, and then generating an example $x$ that satisfies $c$. Thus, we need to design relational programs $c$ that encode constraints on the structure of an example $x$. Our programs $c$ encode a set of *relational constraints*, each of which imposes a constraint that subcomponents of $x$ should have certain kinds of relations. We begin by describing the structure of a single relational constraint, and then describe how $c$ encodes a set of relational constraints.

A *relational constraint* $\phi \in \Phi = \mathcal{W} \times \mathcal{I} \times \mathcal{R}$, where $\mathcal{I} = \{1, ..., m\}$, is a tuple $\phi = (\tilde{w}, i, r)$; we call $\tilde{w} \in \mathcal{W}$ a *prototype subcomponent*. An example $x$ *satisfies* $\phi$ (denoted $x \models \phi$) if $f(\tilde{w}, w_i, r) = 1$, where $w_i$ is the $i$th subcomponent of $x$. That is, $\phi$ says the $i$th subcomponent

$w_i$ of $x$ should have relation $r$ with prototype subcomponent $\tilde{w}$. Thus, we can interpret $\phi$ as a function $\phi : \mathcal{X} \to \mathbb{B}$, where $\phi(x) = 1$ if $x$ satisfies $\phi$ and $\phi(x) = 0$ otherwise.

Next, a relational program $c$ encodes a collection of relational constraints on examples $x$. We represent $c$ as an undirected labeled bipartite graph $c = (\tilde{V}, V, E)$ with vertices $\tilde{V}$ and $V$ and edges $E \subseteq \tilde{V} \times V \times \mathcal{L}$, where $\mathcal{L}$ are the labels. The vertices $\tilde{w} \in \tilde{V}$ are prototype subcomponents $\tilde{w} \in \mathcal{W}$; equivalently, they may be vector embeddings of prototype subcomponents. The vertices $i \in V = \{1, ..., m\}$ are the indices of subcomponents in $x$. The edges $e \in E$ are tuples $e = (\tilde{w}, i, R)$, where $R \subseteq \mathcal{R}$ is a set of relations. We impose the constraint that each $v \in V$ is part of a single edge $(\tilde{w}, v, R)$ (though $\tilde{v} \in \tilde{V}$ may be part of multiple edges). Finally, $c$ encodes the set of relational constraints

$$\Phi_c = \{(\tilde{w}, i, r) \mid (\tilde{w}, i, R) \in E \wedge r \in R\} .$$

In other words, $c$ includes the relational constraint that each subcomponent $w_i$ of $x$ should have all relations $r \in R$ with prototype $\tilde{w}$, where $v$ is connected to $\tilde{w}$.

As an example, in Figure 1, the graph shown on the top right encodes a relational constraint $c_x$, and the top right shows an example $x$ that satisfies all the constraints $\phi \in \Phi_{c_x}$. The nodes on the left-hand side of $c_x$ are prototype subcomponents $\tilde{w} \in \mathcal{W}$, each of which is a line of poetry. The nodes on the right-hand side correspond to indices $i$ (from $i = 1$ on top to $i = m = 10$ on the bottom); each one is labeled with a set of relations $R_i$. Then, $\Phi_{c_x}$ contains constraints $\phi = (\tilde{w}, i, R_i)$ for each edge $\tilde{w} \to i$ in the graph, which says that line $i$ of $x$ should have relations $r \in R_i$ with $\tilde{w}$. For instance, the last (10th) node in $c_x$ has constraints $R_{10} = \{\text{rhyme}, \text{meter}\}$, and is connected to prototype line $\tilde{w} = $ "The harp that thou...". Thus, this edge encodes a constraint $\phi = (\tilde{w}, 10, R_{10})$ saying that the last line of $x$ should rhyme and have the same meter as $\tilde{w}$. Indeed, the last line of $x$ is $w_{10} = $ "Who serve thee most...", which rhymes and has the same meter as "The harp that thou...".

**Remark 3.1.** We use prototypes rather than direct relationships between components to ensure the size of the graph is tractable. In particular, our approach ensures that the graph is linear in the size of the input (assuming the number of prototypes is constant). A compact graph is both to synthesize (for training) or train a model to generate (for generation). Our approach can easily be generalized to more complex representations.

**Remark 3.2.** We refer to $c$ as a program since it can be interpreted as a Datalog program Ceri et al. (1989) (i.e., a relational logic program). At a high level, $\Phi_c$ is a set of Datalog relations over examples $x \in \mathcal{X}$. Thus, $c$ can be interpreted as a program $c : \mathcal{X} \to \mathbb{B}$ such that $c(x) = 1$ if $\phi(x) = 1$ for all $\phi \in \Phi_c$ and $c(x) = 0$ otherwise.

## 3.2 Synthesizing Relational Constraints

Recall that when training our generative model, we need to design a program synthesis algorithm $\mathcal{A}$ that synthesizes a relational program $c_x = \mathcal{A}(x)$ that best encodes the latent relational constraints present in each training example $x$. A key question is where the prototypes come from. We simply choose the prototypes $\tilde{w}$ to be actual subcomponents in $x$. Thus, $c_x$ encodes that subcomponents of $x$ are each related to one of a small number of distinguished subcomponents of $x$. We formulate the problem of synthesizing $c_x$ as a constrained optimization problem, which we describe below.

**Optimization variables.** The variables are a binary vector $H \in \mathbb{B}^m$ and a binary matrix $K \in \mathbb{B}^{m \times m}$. Intuitively, $H_i$ indicates whether subcomponent $w_i$ of $x$ is a prototype subcomponent in $c$, and $K_{ij}$ indicates whether $w_i$ is the prototype for subcomponent $w_j$.

**Constraints.** Our optimization problem has the following three constraints:

$$\psi_1 \equiv k_{\min} \leq \sum_{i=1}^{m} H_i \leq k_{\max}, \qquad \psi_2 \equiv \bigwedge_{j=1}^{m} \sum_{i=1}^{m} K_{ij} = 1, \qquad \psi_3 \equiv \bigwedge_{i=1}^{m} \sum_{j=1}^{m} K_{ij} \leq m \cdot H_i.$$

First, $\psi_1$ says that the number of prototype subcomponents is between $k_{\min}$ and $k_{\max}$. Next, $\psi_2$ says that every subcomponent $w_j$ corresponds to exactly one prototype subcomponent $w_i$. Finally, $\psi_3$ says that for every $i$, if $w_i$ is the prototype subcomponent of $w_j$ according to $K$, then it must be a prototype subcomponent according to $H$ as well.

**Objective.** The objective of our optimization problem is expressed in terms of a precomputed distance matrix $D \in \mathbb{R}^{m \times m}$, where $D_{ij}$ measures the similarity between components $w_i$ and $w_j$; smaller values indicate a greater degree of similarity. In particular, we define

$$D_{ij} = \frac{1}{|\mathcal{R}|} \sum_{r \in \mathcal{R}} \mathbb{1}(f(w_i, w_j, r) = 0),$$

i.e., $D_{ij}$ is the fraction of relations that are not satisfied by $w_i$ and $w_j$. Then, our objective (which is to be minimized) has the following three terms:

$$J_1 = \sum_{i,j=1}^{m} K_{ij} \cdot D_{ij}, \qquad J_2 = \sum_{i,j=1}^{m} \left( \prod_k K_{ki} \cdot K_{kj} \right) \cdot D_{ij}, \qquad J_3 = - \sum_{i,j=1}^{m} M_i \cdot M_j \cdot D_{ij}.$$

First, $J_1$ says that subcomponents should be similar to their prototypes. Second, $J_2$ says that subcomponents should also be similar to other subcomponents that share the same prototype. Third, $J_3$ says that different prototype subcomponents should be dissimilar.

**Optimization problem.** Our algorithm $\mathcal{A}$ uses Z3 to solve the optimization problem

$$(H^*, K^*) = \underset{H,K}{\arg\min} \{\lambda_1 \cdot J_1 + \lambda_2 \cdot J_2 + \lambda_3 \cdot J_3\} \qquad \text{subj. to} \qquad \psi_1 \wedge \psi_2 \wedge \psi_3,$$

where $\lambda_1, \lambda_2, \lambda_3 \in \mathbb{R}_{\geq 0}$ are hyperparameters. Finally, to construct $c_x$, $\mathcal{A}$ chooses

$$\tilde{V} = \{w_i \mid H_i^* = 1\}, \qquad V = \{1, ..., m\}, \qquad E = \{(w_i, j, R_{ij}) \mid K_{ij}^* = 1\},$$

where $R_{ij} = \{r \in \mathcal{R} \mid f(w_i, w_j, r) = 1\}$—i.e., $\tilde{V}$ are the prototype subcomponents according to $H^*$, $E$ are the edges according to $K^*$, and $R_{ij}$ are the relations satisfied by $w_i$ and $w_j$.

Z3 is guaranteed to find the optimal solution; in the unlikely event that multiple such solutions exist, it chooses one nondeterministically. Intuitively, our approach should perform well when a handful of prototypes are sufficient to approximately capture the relational structure in the data. Furthermore, since the user has the ability to define relations, they can adjust their definitions as needed to capture the desired structures.

## 4 Neurosymbolic Generative Models with Relational Constraints

In this section, we describe our deep generative model for generating examples $x$. Recall that our approach proceeds in two steps: (i) generate $c$, and (ii) generate $x$ given $\Phi_c$. We describe each of these steps in detail below.

### 4.1 Step 1: Generating Relational Constraints

The first step of our generative model is to generate relational constraints $\Phi_c$ using a VAE—i.e., $z \sim p(\cdot)$ and $c \sim p_\phi(\cdot \mid z)$, where $p_\phi(c \mid z)$ is a VAE and $p(z) = \mathcal{N}(z; 0, I)$ is a Gaussian distribution. The main choice is the architecture to use for the VAE. In particular, we consider a representation of $c$ as a sequence $(s_1, ..., s_m)$, where $s_i \in \{0, 1, ..., m\}$ for each $i$; intuitively, $s_i$ encodes that subcomponent $w_i$ should have the same prototype subcomponent as $w_{i-s_i}$, or if $s_i = 0$, that $w_i$ corresponds to a new prototype subcomponent.

More precisely, we initialize $\Phi_c = \varnothing$. Then, we generate the sequence $s_i \in \{0, 1, ..., m\}$ and $r_i \in \{0, 1, ..., m\}$ (where $r_i$ is represented as a binary vector of length $n = |\mathcal{R}|$) using an LSTM-VAE. For each $i$, we generate $(\tilde{w}, R_i)$ based on $s_i$ and $r_i$. If $s_i = 0$, we generate a new prototype subcomponent $\tilde{w}$ using a domain-specific generative model, generate with using another LSTM-VAE, and add $\phi_i = (\tilde{w}_i, i, R_i)$ to $\Phi_c$. If $s_i > 0$, we let $\phi_i = (\tilde{w}_{i-s_i}, i, R_i)$.

### 4.2 Step 2: Generating Examples Given Relational Constraints

Next, we describe how we implement the second step $p_\theta(x \mid c)$ of our generative model. We propose three approaches for generating $x$ given $\Phi_c$; we give details in Appendix A.

**Approach 1: Constrained sampling.** We sample values $x \sim p_\theta(\cdot)$ by sequentially sampling $w_i \sim p_\theta(\cdot)$ from a pretrained generative model $p_\theta(w)$. We do so using rejection sampling at each step—i.e., we sample $w_i \sim p_\theta(\cdot)$ until we find $w_i$ satisfying $f(\tilde{w}, w_i, r) = 1$ for each $(\tilde{w}, i, r) \in \Phi_c$. In addition, to speed up sampling, at each step of sampling $w_i$ (e.g., a word in a line or a pitch in a measure), we eliminate choices that violate $\Phi_c$.

**Approach 2: Constraint-aware embeddings.** We train a conditional generative model $p_\theta(w_1, ..., w_m \mid c)$ (in the form of a graph convolutional network) that simultaneously generates all $m$ subcomponents in a way that satisfies $c$, and sample $x = (w_1, ..., w_m) \sim p_\theta(\cdot \mid c)$.

**Approach 3: Combinatorial optimization.** We sample $x \sim p_\theta(\cdot)$ by sequentially generating $w_i$ by solving an optimization problem whose objective is to maximize adherence to $\Phi_c$ plus additional terms encoding domain-specific heuristics encouraging $w_i$ to be realistic.

## 5 Experiments

We evaluate our approach on music and poetry generation; see Appendix B.6 for details.

**Music generation.** We evaluated our approach on a music generation task. In this setting, $x$ is a song, and $w$ are measures of music. We consider 20 relations including equality, same rhythm, same pitch progression, etc.; a full list is given in Appendix B.2. We used songs from the Essen folk song corpus (Schaffrath, 1995), using 2000 for training and 500 for testing. For this dataset, we used each of the three approaches A1, A2, and A3 described in Section 4 to sample $x \sim p_\theta(\cdot \mid c)$. For A1, we use a pretrained transformer called MusicAutoBot (Shaw, 2020). For A2, we require a generative model that constructs vector embeddings of measures; we use the version of Magenta's MusicVAE which embeds single measures (Roberts et al., 2018). We finetune all models on our training examples.

We compare to MusicAutoBot, an LSTM model with attention (AttentionRNN) (Waite, 2016), Magenta's 16-bar MusicVAE, and an implementation of StructureNet, an approach that integrates structure into an LSTM (Medeot et al., 2018). We also compare to a constraint generation approach called Motifate (Muhammad Faisal, 2017); see Appendix B.5. We compare performance in terms of both high-level and low-level structure. For high-level structure, given a generated (or human held-out) example $x$, we use our program synthesis algorithm to synthesize its relational structure $c_x = \mathcal{A}(x)$. Then, given a collection $C_{\text{gen}} = \{c_x \mid x \in X_{\text{gen}}\}$ of synthesized structure for generated examples, along with a collection $C_{\text{human}} = \{c_x \mid x \in X_{\text{human}}\}$ of synthesized structure for the held-out human examples, we train a graph convolutional neural network (GCN) to try and discriminate $C_{\text{gen}}$ from $C_{\text{human}}$. In addition, we also train a random forest (RF) over handcrafted features (described in B.4) to try and discriminate them. In both cases, we use a balanced dataset (i.e., 50% human held-out and 50% generated) so random predictions have accuracy 0.5. For low-level structure, we use the negative log likelihood (NLL) according to MusicAutoBot (pretrained and then fine-tuned on our dataset), MusicVAE, and our own GraphVAE (described in Section A.2). While these metrics are not perfect, they can be used to evaluate across all approaches. For models where NLL is available, we also compare their NLL on the held-out humand data.

We show results in Table 1. For each approach (as well as a held-out human dataset), we show the negative log-likelihood (NLL) assigned to that approach by one of the three models "MusicAutoBot", "MusicVAE", and "GraphVAE". According to almost all metrics, our algorithm (using A2 for sampling $p_\theta(x \mid c)$) outperforms or performs equally to all others, and achieves performance very similar to human. The one exception is our measure of low-level structure according to MusicAutoBot; however, this model rates StructureNet and MusicVAE-16 as substantially better than human, indicating that it is not a good measure of quality. Finally, our GraphVAE produces a lower NLL for the held-out human music than either MusicAutoBot or MusicVAE-16, which indicates that it models human data better than the others (the other approaches cannot be used to compute NLL).

**Poetry generation.** We use Project Gutenberg's poetry collection (Parrish, 2018), filtered to focus on examples that contain rhymes and meter. We use 2700 10-line poems for training and 300 for testing. In this case, we were unable to apply A3 due to the large

| Model | Low-Level | | | High-Level | |
|---|---|---|---|---|---|
| | MusicAutoBot | MusicVAE | GraphVAE | RF Disc. | GCN Disc. |
| Ours (A1) | 1518 | 1161 | 1037 | 0.89 | 0.54 |
| Ours (A2) | 1184 | **1156** | **1027** | **0.79** | **0.63** |
| Ours (A3) | 1240 | 1161 | 1029 | 0.91 | 0.43 |
| MusicVAE-16 | 1093 | 1170 | 1070 | 0.85 | 0.50 |
| MusicAutoBot | 5452 | 1172 | 1100 | 0.95 | 0.51 |
| AttentionRNN | 1920 | 1271 | 1089 | 0.88 | 0.47 |
| StructureNet | **902** | 1162 | 1062 | 0.91 | 0.45 |
| Human | 1764 | 1160 | **1028** | *0.51* | *0.69* |

Table 1: Results for the music domain. To evaluate low-level structure, we use negative log likelihood according to MusicAutoBot, Magenta's verrsion of MusicVAE designed for hierarchical 16-bar melodies, and GraphVAE. For high-level structure, we use accuracy of the random forest ("RF Disc.") and GCN cross entropy loss ("GCN Disc."). The best (non-human) score in each column is bolded; the human score is italicized if best. We also bold the model that achieves the best NLL on the held-out human data.

| Model | Low-Level | | High-Level | Diversity |
|---|---|---|---|---|
| | BERT | GPT2 | GCN Disc. | Entropy |
| Ours (A1) | 3.66 | 6.51 | **0.69** | 4.64 |
| Ours (Ablation) | 3.72 | 6.60 | 0.59 | 4.67 |
| GPT2 | **3.30** | **3.13** | 0.47 | 4.32 |
| GPT2-Opt | 3.63 | 3.87 | 0.56 | 4.35 |
| BERT | 3.90 | 5.76 | 0.50 | 2.59 |
| RichLyrics | 4.66 | 8.04 | 0.51 | **4.86** |
| Human | 4.02 | 4.98 | *0.70* | *4.86* |

Table 2: Results for the poetry domain. To evaluate low-level structure, we use the negative log likelihood per token of a fine-tuned version of BERT and GPT2. For high-level structure, we use cross-entropy loss of the GCN ("GCN Disc."). We also show the information entropy. The highest (non-human) score in each column is bolded.

size of the vocabulary, making constrained optimization infeasible. We were also unable to apply A2 since state-of-the-art generative models such as BERT and GPT2 were unable to capture rhyming and meter, since they operate at the word level where this information is unavailable. In A1, rather than sample words going forward, we sample them backwards, making it easier to sample lines that satisfy rhyming constraints; see Appendix A. Thus, we use BERT to sample (Devlin et al., 2018), since it supports bidirectional sampling.

We compare to BERT and GPT2 (Radford et al., 2019), both finetuned on our dataset. We also consider a variant GPT2-Opt of GPT2 where we use beam search to choose line breaks in a way that maximizes occurrences of rhyme and meter. We also tried a variant of GPT2 that used constrained sampling to try and find poems that fit a given rhyme and meter scheme, but the search space was too large and it was unable to generate a single poem even after several hours. We also compare to an implementation of RichLyrics (Castro & Attarian, 2018), where the consecutive parts of speech for each line given the previous line and the ability to fill in the correct word for the given part of speech were both learned separately from the corpus. Finally, in addition to using BERT as a sequential generator, we considered an ablation where we perform constrained sampling, but with a uniformly random $\Phi_c$ rather than sampling it from a learned distribution.

As before, we compare both high-level structure and low-level structure. For high-level structure, we again use a GCN discriminator. For low-level structure, we use the negative log likelihoods per token according to each of BERT and GPT2 (finetuned on our training dataset). In this case, because our approach uses constrained sampling from a pretrained generative model, we could not evaluate negative log-likelihood according to our approach.

We show results in Table 2. Our approach significantly outperforms all baselines (including RichLyrics) in terms of high-level structure. Furthermore, our approach produces a lower BERT NLL than either unmodified-BERT or the BERT-based implementation of Rich-Lyrics. GPT2 and GPT2-Opt produce more likely output than our technique according to

One was done. Another was done.
And I wish you know the way,
Full name and date to whom this story pour
And know a lot of things that were called a war
See a soldier, fair fair beautiful grace
That men turn'd toward. Another race
Together, married. Much to see, the dead
Were gone. The man who ascended to the head
Office retired, and gave birth to a trace
That doesn't tell a name, but tells a face.

One was done. Another was done.
And I wish you know the way,
Full name and date to whom this story pour
And know a lot of things that were called a war
See a soldier, fair fair beautiful grace
That men turn'd toward. Another race
Together, married. Much to see, the dead
Were gone. The man who ascended to the head
With full beard and hair was a little said
But was old and not intended for bed.

Of nature and of nature nature is the
Only being able in human affairs to
Combine with herself
Her will and therefore her existence cannot ever fail even
As nature having no desire can create itself so too
Alone can nature produce any being the
Human existence cannot
Then exist because
It only can
Exist because the nature only is

Figure 2: Left: Poetry generated using relational constraints $c \sim p_\phi(\cdot)$. Middle: user modified variant of $c$ where the last two lines share a prototype with the two lines before them. Right: A poem generated by GPT2 optimized to maximize rhyme and meter. The colors indicate the relations synthesized by our algorithm *after* the examples were generated.

the transformer probability metrics, most likely because they are better at natural language generation than BERT. If we could instead build on GPT2 (i.e., perform backwards sampling with GPT2), then our approach would likely achieve better performance; we leave this direction to future work. Furthermore, our approach achieved comparable BERT scores to GPT2-Opt, while significantly outperforming it in terms of structure.

Finally, we noticed that one way the baselines (except for the ablation study and RichLyrics) tended to perform well was by being very repetitive. Thus, we additionally measured the information entropy of the different models. As can be seen, our approach (both with learned and random $\Phi_c$) is by far closer to human in terms of human entropy than GPT2-Opt and BERT, supporting our hypothesis that the baselines were overly repetitive. RichLyrics avoided this lack of entropy, as consecutive words were constrained to belong to different part of speech groups. However, it did not receive a high likelihood score, as the transformer model used to produce those parts of speech often resulted in unlikely output.

In Figure 2, we show an example poem generated using our approach (left) along with one generated using GPT2-Opt (right). As can be seen, the GPT2-Opt poem does not capture structure in the same way human poems do—e.g., adjacent lines are unrelated, lines have very unequal length, and the only rhymes are the word "the" in the brown lines and the words "to" and "too" in the green lines. There is even less structure in poems generated using vanilla GPT2. Thus, GPT2 is completely unable to capture high-level structure present in the real poetry provided as training data. In contrast, our poem captures structure very similar to the human poem shown in Figure 1, such as rhyming adjacent lines.

**User modifications.** A key benefit of our approach is that the user can modify the relational constraints $c$ (or construct their own from scratch) for use in the second step $p_\theta(x \mid c)$, giving the user a way to guide the generative process. Figure 2 demonstrates this process. We manually edited the part of the program corresponding to the last two lines so that they shared a prototype with the previous two lines. The example generated using the unmodified (sampled) constraints is shown on the left, and the example generated using the modified constraints is shown in the middle. We show in the supplement a similar process performed with music data.

## 6 CONCLUSION

We have presented a novel approach for representing and synthesizing relational constraints on sequence data, and for generating examples whose relational structure resembles that of the training data. Our experiments demonstrate that we outperform existing approaches in terms of achieving human-like structure, while performing comparably or better on widely-used metrics which do not explicitly account for structure. Equally importantly, our approach gives the user a way to guide the generative process by modifying the relational constraints. Directions for future work include automatically discovering relational primitives, integrating our structure generator with more powerful language models such as GPT-3, and improving the ability to sample from generative models subject to constraints.

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

## A  Generating Examples Given Relational Constraints

### A.1  Approach 1: Constrained Sampling

In the music domain, we choose the pretrained generative model $p_\theta(w)$ to be a pretrained version of MusicAutoBot. To generate $x$, we sequentially sample each measure $w_i$ conditioned on all prior measures $w_1, ..., w_{i-1}$. Each measure is sampled by sequentially sampling a sequence of pitch-duration pairs until the total duration is 16 beats (i.e., the length of a measure). During sampling, we mask pitch-duration pairs that cannot satisfy $\Phi_c$ (i.e., we set their sampling probability to zero and rescale the remaining probabilities). For instance, if the "has similar interval" relation is supposed to hold between the the prototype measure and measure $i$, and we are sampling the second note of measure $i$, then we mask any pitch $j$ in measure $i$ such that

$$|(\text{pitch}_j - \text{pitch}_{j-1}) - (\widetilde{\text{pitch}}_j - \widetilde{\text{pitch}}_{j-1})| \geq 3,$$

where $\widetilde{\text{pitch}}_j$ is pitch $j$ in the prototype corresponding to $w_i$. In other words, we eliminate pitches that would cause sampling to violate this constraint.

In the the poetry domain, we finetune a pretrained BERT model on our dataset, by taking the pretrained models weights and then training the model on our dataset with a strong gradient weight decay. BERT has the ability to complete masked words in a sentence. We leverage this ability to sample lines that rhyme and have the same meter, which is a challenging task since such lines are a tiny fraction of the search space. We describe how we simultaneously handle rhyming and equal meter; the cases where only one of these two constraints has to hold are similar. Given a prototype $\tilde{w}$, we work backwards—on each step $j$, we sample from BERT a word $\text{word}_j$ that has the same number of syllables as the corresponding word $\widetilde{\text{word}}_j$ in the prototype. More precisely, we feed BERT the sequence

$$\widetilde{\text{word}}_1, ..., \widetilde{\text{word}}_{j-1}, \text{MASK}, \text{word}_{j+1}, ...$$

and ask it to fill in the masked word, setting the probability of any word with different number of syllables as $\widetilde{\text{word}}_j$ to zero. In addition, we also set the probability of any word too similar to the original word in terms of cosine similarity to zero. For the last word (which we sample first), we additionally restrict to words that rhyme with $\widetilde{\text{word}}_j$. To increase diversity, we sample the remaining words twice—(i) backwards-to-forwards from word $k-1$ to word 1, where $k$ is the number of words, and (ii) we resample each of the $k-1$ words (i.e., except the last word) in a random order. We discard any lines which after being sampled are determined to be too unlikely according to BERT.

### A.2  Approach 2: Constraint-Aware Embeddings

In this approach, we start with a pretrained generative model $p_\theta(w)$ that ignores $c$; in particular, we assume that

$$p_\theta(w) = \int p_\theta(w \mid u) \cdot p(u) du,$$

where $p_\theta(w \mid u)$ is the decoder network of a VAE over $w$, and $p(u) = \mathcal{N}(0, I)$. Now, rather than sample $w_i \sim p_\theta(\cdot)$, we train another generative model

$$p_\psi(u_1, ..., u_m ... u_m \mid c) = \mathcal{N}((u_1, ..., u_m) \mid \mu_\psi(c), \Sigma_\psi(c))$$

to generate latent vectors $u_i \in \mathcal{U}$ such that $w_i \sim p_\theta(\cdot \mid u_i)$ are likely to satisfy $\Phi_c$. More precisely, $\mu_\psi$ and $\Sigma_\psi$ are the intermediate outputs of a graph convolutional network (GCN) (Kipf & Welling, 2017) that takes as input the graph $c$ (where edge attributes between nodes encode $\Phi_c$) and ultimately outputs a sequence $(u_1, ..., u_m)$.

Our approach can be considered to be a graph autoencoder in the sense that the objective function used during training rewards the reconstruction of the exact embeddings of the nodes and (implicitly, through the relationship consistency loss) their edge attribute. Our

graph encoder/decoder produces one latent vector per node, which are rewarded for close to i.i.d. Gaussian random variables with mean zero and variance one.

To train $p_\psi$, we construct a training example $(c_x, (u_1, ..., u_m))$ for each training example $x = (w_1, ..., w_m)$, where $u_1, ..., u_m$ are obtained by the encoder network $q_{\tilde\theta}(u \mid w)$—i.e., $u_i \sim q_{\tilde\theta}(\cdot \mid w_i)$. Then, we train $p_\psi$ using the objective

$$J(\psi) = \sum_{(c,\vec u)} D_{\mathrm{KL}}(\mathcal{N}(\mu_\psi(c), \Sigma_\psi(c)) \parallel \mathcal{N}(0, I)) + \sum_{i=1}^{m} \|u_i - \mu_\psi(c)_i\|_2^2 + J_{\mathrm{rel}}(\mu_\psi(c); c).$$

The first term enforces that the distribution of the latent vectors $u$ should be Gaussian, and the second term enforces that each latent vector $u$ should be close to its original value according to the VAE encoder $q_{\tilde\theta}(u \mid w)$. The third term is designed to enforce the satisfaction of the constraints $\Phi_c$. In particular, we train a kind of "semantic discriminator" $p_\alpha(u, u'; r)$, that predicts whether $w \sim p_\theta(\cdot \mid u)$ and $w' \sim p_\theta(\cdot \mid u')$ satisfies relation $r$— i.e., $f(w, w', r) = 1$. The network $p_\alpha$ is trained on data generated from the given training examples $x$. Then, given $p_\alpha$, we want $(u_1, ..., u_m) = \mu_\psi(c)$ to satisfy

$$p_\alpha(u_i, \tilde u, r) \approx \begin{cases} 1 & \text{if } (\tilde w, i, R) \in \Phi_c \wedge r \in R \\ 0 & \text{otherwise} \end{cases} \qquad \text{where} \qquad \tilde u \sim q_{\tilde\theta}(\cdot \mid \tilde w).$$

In other words, we want to generate an example $x$ that satisfies the relations in $c$ according to $p_\alpha$. In particular, we use the loss

$$J_{\mathrm{rel}}(\vec u; c) = \sum_{i=1}^{m} \sum_{r \in \mathcal{R}} \mathrm{CE}\Big(p_\alpha(u_i, \tilde u, r), \mathbb{1}\big((\tilde w, i, R) \in \Phi_c \wedge r \in R\big)\Big) \qquad \text{where} \qquad \tilde u \sim q_{\tilde\theta}(\cdot \mid \tilde w),$$

and where CE denotes the cross-entropy loss. Once we have trained $p_\psi$, we generate sequences by sampling $\vec u \sim p_\psi(\cdot \mid c)$ and $w_i \sim p_\theta(\cdot \mid u_i)$, and constructing $x = (w_1, ..., w_m)$.

For the music domain, we use embeddings from a pretrained Magenta MusicVAE; unlike the MusicVAE used for evaluation, we finetuned it to decode only 1-2 measures of music from a 256-dimensional vector. Then,w e use the decoder portion of this model to convert the embeddings $u_1, ..., u_m \sim p_\psi(u_1, ..., u_m \mid c)$ sampled from the GCN-VAE $p_\psi$ into measures. The graphs in the training set vary in size depending on the number of prototype measures.

## A.3   APPROACH 3: COMBINATORIAL OPTIMIZATION

Given sampled program $c$, this approach attempts to generate values $x = (w_0, \ldots, w_m)$ such that $x \models \Phi_c$ by solving a system of constraint solving problem. However, the relational constraints $\phi \in \Phi_c$ are not always consistent with one another, so we relax the constraint $x \models \Phi_c$ as an objective—i.e.,

$$x = \arg\max_{x \in \mathcal{X}} \sum_{i=1}^{m} \sum_{r \in \mathcal{R}} \mathbb{1}(\mathcal{R}(\tilde w, w_i, n) \Leftrightarrow (\tilde w, i, n) \in \Phi_c).$$

Encoding this optimization problem as one Z3 can solve depends on the domain and relations. For this approach to work, we need to include additional, handcrafted terms in the objective that encourage the generated example $x$ is realistic.

For the music domain, the optimization variables are the optimal sequence of pitches and their durations. The objective function is a linear combination of the degree to which $x$ satisfies $c$, along with domain-specific heuristics—e.g., minimizing large jumps in pitch values (i.e., $|\mathrm{pitch}_{i+1} - \mathrm{pitch}_i| \geq 4$), not having any intervals of length 6 (i.e., $|\mathrm{pitch}_{i+1} - \mathrm{pitch}_i| = 6$) due to the unpleasant harmonic nature of that interval, and not having two consecutive jumps in pitch (i.e., $|\mathrm{pitch}_{i+2} - \mathrm{pitch}_{i+1}| \geq 5) \wedge |\mathrm{pitch}_{i+1} - \mathrm{pitch}_i| \geq 5$). These heuristics are based on standard concepts from music theory (Horton & Ritchey, 2000).

# B  EVALUATION DETAILS

## B.1  EXPERIMENTAL SETUP

**Generating** $c$**.** To generate $c$, we use an LSTM-VAE with 2 LSTM layers and a latent size of 200 in the music domain and 50 in the poetry domain. This model is trained to reproduce a given sequence of $(s_i, r_i)$ pairs which are given as input, with an additional requirement that the distribution of their encodings should be roughly equivalent to a Gaussian normal distribution. Each $(s_i, r_i)$ pair is represented as a $(S + |R|)$-dimensional vector, where $S$ is the maximum distance between objects with the same prototype and $R$ is the set of relations.

**Evaluating low-level structure.** We evaluate low-level structure by using negative log likelihood according to a deep generative model. In the music domain, we use both MusicAutoBot, a transformer based on the transformer-XL architecture (Dai et al., 2019), as well as the MusicVAE from Magenta, which is a hierarchical VAE that learns embeddings for each measure and then learns an LSTM-VAE on top of these embeddings. In the poetry domain, we use BERT and GPT2, both finetuned on our dataset and solely pretrained on a non-poetry dataset. For the evaluation metrics that explicitly captured structure, we computed the optimal program $\mathcal{A}(x)$ for every example $x$ in the held-out validation human dataset as well as all of the generated examples.

**High-level structure.** We evaluate high-level structure by using our algorithm to synthesize the relational constraints in every generated example—i.e., $C_{\text{gen}} = \{\mathcal{A}(x) \mid x \in X_{\text{gen}}\}$, where $X_{\text{gen}}$ is the set of examples generated using a model. Similarly, we can construct $C_{\text{human}} = \{\mathcal{A}(x) \mid x \in X_{\text{human}}\}$, where $X_{\text{human}}$ is the set of human-created examples held-out from the training dataset. Then, we evaluate high-level structure by training a model to try and discriminate $C_{\text{gen}}$ from $C_{\text{human}}$; if the model achieves lower performance, then the quality of high-level structure is higher. A general approach is to train a graph neural network (e.g., a graph convolutional network) to do so; this model takes as input the graph structure of relational constraints $c$, along with vector embeddings of the prototype subcomponents, and outputs whether $c \in C_{\text{gen}}$ or $c \in C_{\text{human}}$. We balance the data so it consists of 50% human data and 50% generated data. We report the cross-entropy (CE) loss; higher values correspond to better generative models. In the music domain, we additionally used a random forest (RF) trained on a manual featurization of $c$. We report the accuracy of the RF; lower values (i.e., closer to 50%) correspond to better generative models.

## B.2  MUSICAL RELATIONS USED

The following are the relations $r \in \mathcal{R}$ used in the music domain:

1. Measures $i$ and $j$ have the same pitch classes.
2. Measures $i$ and $j$ have the same pitch class prefix.
3. Measures $i$ and $j$ have the same pitch class suffix.
4. Measures $i$ and $j$'s pitches have an edit distance of 1.
5. Measures $i$ and $j$ have approximately the same interval structure.
6. Measures $i$ and $j$ have the same interval prefix.
7. Measures $i$ and $j$ have the same interval suffix.
8. Measures $i$ and $j$ have the same note (pitch + duration) prefix.
9. Measures $i$ and $j$ have the same note (pitch + duration) suffix.
10. Measures $i$ and $j$ have the same rhythm.
11. Measures $i$ and $j$'s rhythm has an edit distance of $\leq 2$.
12. Either measure $i$'s onsets are a subset of measure $j$'s onsets, or measure $j$'s onsets are a subset of measure $i$'s onsets.
13. Measures $i$ and $j$ have the same rhythmic and melodic contour.

14. Measures $i$ and $j$ have the same rhythmic and melodic contour prefix.

15. Measures $i$ and $j$ have the same rhythmic and melodic contour suffix.

16. Either the first or second half of measures $i$ and $j$ are identical.

17. Either both or neither of measures $i$ and $j$ have leaps.

18. Measures $i$ and $j$ fit within the same diatonic scale.

19. Either both or neither of measures $i$ and $j$ have syncopation.

20. Either both or neither of measures $i$ and $j$ have consecutive notes shorter than an eighth note.

### B.3 Poetry Relations Used

The following are the relations $r \in \mathcal{R}$ used in the poetry domain:

1. Lines $i$ and $j$ have the same end rhyme.

2. Lines $i$ and $j$ have the same meter.

### B.4 Random Forest Features

The following are the manually constructed features used in the random forest discriminator for the music domain:

1. Mean number of relations between prototype and sequence measures.

2. Variance of number of relations between prototype and sequence. measures

3. Variance in histogram of prototype measure mappings.

4. Longest sequence $i \ldots j$ such that $w_i \ldots w_j$ all have the same prototype measure.

5. Number of pairs $(i, j)$ such that $\tilde{w}_i = \tilde{w}_j$ and $\tilde{w}_{i+1} = \tilde{w}_{j+1}$.

6. Mean distance between two measures with the same prototype.

7. Variance in distance between two measures with the same prototype.

### B.5 Comparison to Constraint Solving

We also considered a comparison to a constraint-based implementation called Motifate, with explicit attention to development of musical material (Muhammad Faisal, 2017). This approach was designed with heuristics for 3-beat measures, while our evaluation models anticipated 4-beat measures, so we could not obtain NLL scores. Nevertheless, we found that even the structure was insufficient—our RF discriminator had accuracy 0.91, and our GCN discriminator had cross entropy loss 0.43, both of which are significantly worse than the other approaches.

### B.6 Qualitative Results

**Music domain.** In addition to quantitative measurements, we evaluated the strengths and weaknesses of our approach using A2 (which was the best according to quantitative metrics). According to our observations, the strengths of A2 include clearer phrases with obvious resolutions, likely and plausibly repetitive rhythms, intervals between notes which seemed plausible but not overly repetitive, and less variance in quality. However, the results were not very rhythmically diverse, and certain idiomatic patterns of resolutions of intervals between notes and at the end of phrases were not followed. Furthermore, AttentionRNN does better in terms of creating realistic chord progressions (we did not explicitly consider chord progressions in our model; doing so is a promising direction for future work). Finally, while global structure is much better than the baselines, examples still relatively infrequently had the full four-bar repetitions characteristic of much folk music.

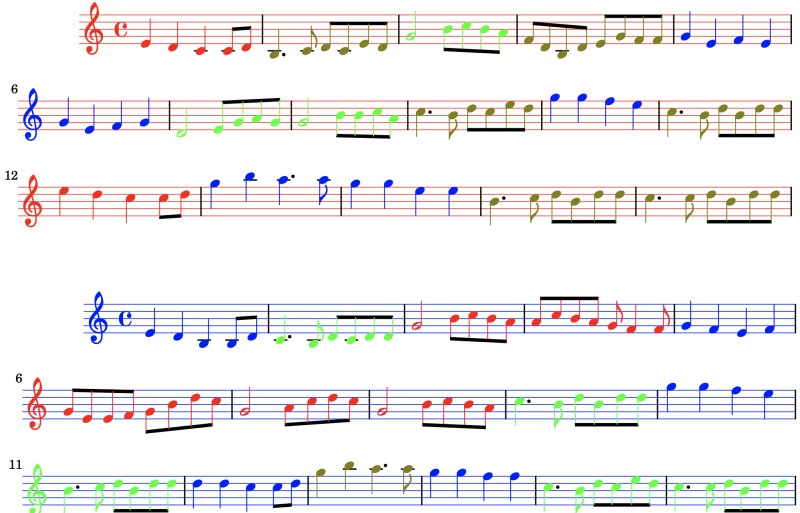

Figure 3: A song generated using our approach A3 (top), and a nearly identical song generated where part of the sampled relational constraints $c$ were manually modified. These pieces were generated using A3, and the same reference measures $\tilde{w}$ were used, but $\Phi_c$ was slightly perturbed (the similarity relations were changed).

**Examples.** Here we show how user modifications can occur in the music setting. By explicitly modifying $c$, we are able to generate two pieces with similar internal patterns but with different structural characteristics.

We show an example of generated songs using our approach with each A1, A2, and A3 in Figure 4, Figure 5, and Figure 6, respectively, and show an example generated using each of the baselines MusicVAE16, AttentionRNN, MusicAutoBot, and Structurenet in Figures 7, 8, 9, and 10, respectively. Qualitatively, the generated music and poetry appears plausible, exhibiting realistic high-level structure without sacrificing low-level structure.

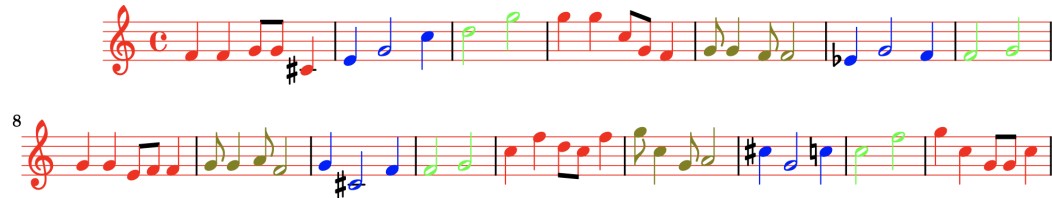

Figure 4: An example of a song generated using our approach (A1). Measures that have the same prototype are shown in the same color. Note the existence of repeating four-bar phrases, found commonly in folk songs.

We also give examples of poetry generated using our baselines—in particular, GPT2 fine-tuned and optimized for rhyme and meter in Figure 12, BERT finetuned as a language generation model in Figure 14, RichLyrics, and our ablation (i.e., use BERT in conjunction with a uniformly randomly sampled $\Phi_c$) in Figure 15.

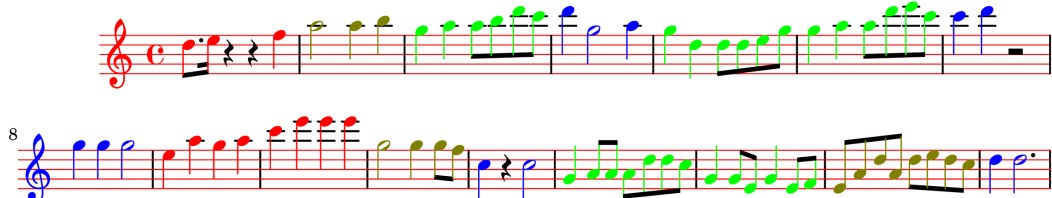

Figure 5: An example of a song generated using our approach (A2). Measures that have the same prototype are shown in the same color. Note the existence of clear phrase endings marked by long notes or rests, particularly the recurring pattern of fast notes resolving into long notes.

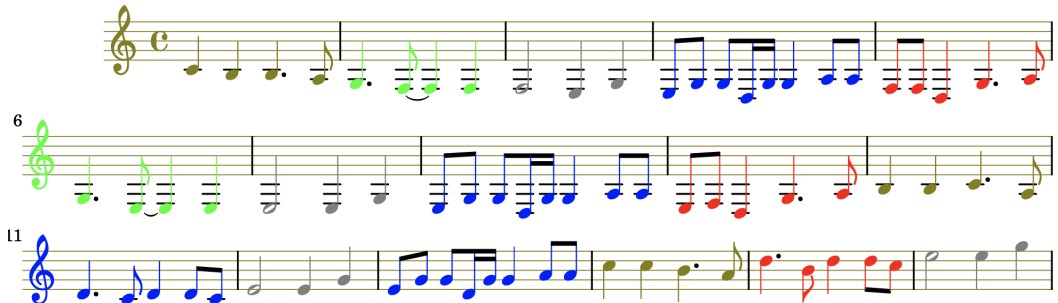

Figure 6: An example of a song generated using our approach (A3). Measures that have the same prototype are shown in the same color. The existence of two-bar and three-bar phrases is apparent, but the close note and rhythm similarities among different prototypes weaken the overall clarity of the song's melody.

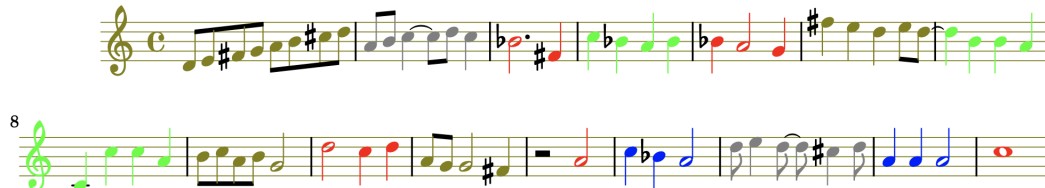

Figure 7: An example of a song generated using Magenta's hierarchical MusicVAE model finetuned on our dataset. While the local structure is extremely coherent, it does not seem to possess the expected internal repetition/development.

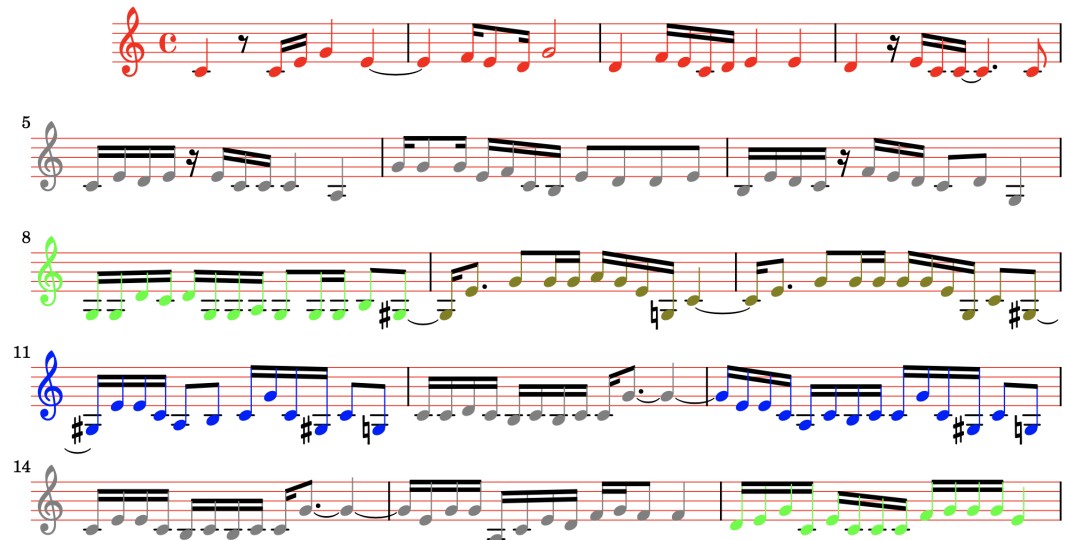

Figure 8: An example of a song generated using AttentionRNN trained on our dataset. Note the existence of erratic rhythms and unclear structure, which are common traits of custom-trained AttentionRNN models.

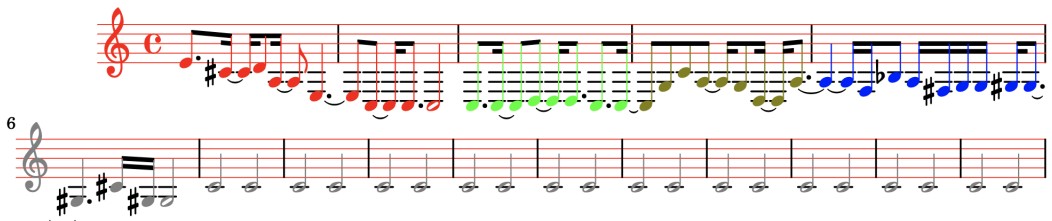

Figure 9: An example of a song generated using MusicAutoBot. Note the repetitive nature and stark contrast between the first half and second half of the song, which are common problems with transformer models.

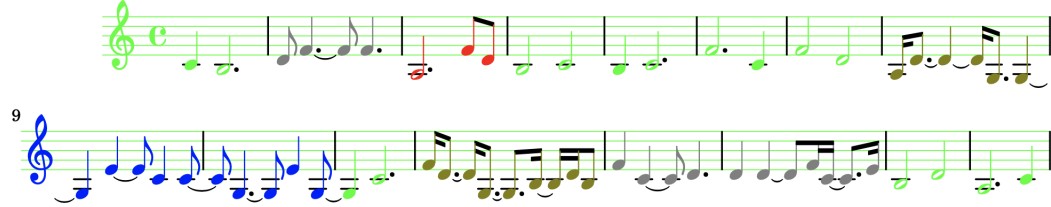

Figure 10: An example of a song generated using StructureNet. While some degree of internal structure is apparent, and the local coherence is high, the pattern of internal repetition seems fairly arbitrary.

I know many things, and therefore I forgot,
Though I needed time to look ahead,
To understand something, time to let it fade away
As though it was yesterday as they
Were common things, free, rather—free, to go like the tide;
But another is to make no one, as it does.
Perhaps you know it. A queen, her beautiful son,
And another woman who has to go without one.
The voices like their cries of war,
They let us believe in a good restore!

Figure 11: An example of poetry generated using our approach. Lines that have the same prototype are shown in the same color.

Through the air and through the sky
And through all the world
I saw the sun the moon a star shine
In the midst of the stars
The stars were shining in my eyes my heart
Was throbbing with joy I felt
My heart was beating with love
I was
A little child in a little town
Where the little boys play

Figure 12: A poem generated using GPT2-Opt. It is more plausible than BERT in terms of of global structure, which may be due to the fact that GPT2 is a better text generation tool than BERT, but it is still somewhat repetitive and its structure is not very human-like.

all all and and
and and and
and and all
all all all
and and o
and and and
o and and
o o and o and
and of of of and o o o but and and of
and and a and and last last last of of and and

Figure 13: A poem generated using BERT. It is clearly overly repetitive and not very semantically coherent, and lacks high-level structure.

and after all text that all more appointment
make its room from sat and all district without self
she one is hundred first enjoy her
but her been two you shall be one
above she leave enjoying the suffering usual
for which more science this day sewing
you shall two houses for recent contributions
and time which have left for self woman
and all moreover let use been found called
and might where out any boots not accident

Figure 14: A poem generated using RichLyrics. While it is less repetitive than non-conditioned BERT, it is still not very semantically coherent, and lacks high-level structure.

the first - independent , like for
the new songs for morning ,
a little world , they asked them for a way .
she asked them for a night ,
with two beds but sometimes lying on a light -
bed , the first for women , with another , one band ,
with one paul simon never got a play
on the subject , she ' d bought
a different dress for a different tent ,
and one dress for warning . .

Figure 15: A poem generated using our ablation. While it is much more coherent, it lacks the idiomatic rhyme and meter structure of our approach.

