# OpenReview forum: "Neurosymbolic Deep Generative Models for Sequence Data with Relational Constraints"
_ICLR.cc/2021/Conference — Reject_

### Official Review · AnonReviewer3 · 2020-10-22
**Interesting method, but insufficient comparison to existing methods and analysis of results**

**Rating:** 4
**Confidence:** 4

**Review:**

**Summary**

This paper presents a novel method of incorporating structural and relational constraints into the generative process of sequences. This approach is explored in both the music and poetry domains using several generative approaches. An analysis of the resulting sequences finds that they successfully incorporate structure.

**Strengths**

Novel procedure for extracting constraints from training data

Results explored in multiple domains

Example of interactive user modifications, showing that the constraints are human-interpretable and useful for guiding generation toward a creative goal

**Concerns**
My first big concern is a lack of comparison against other methods for using structural constraints in the generative process. A couple examples:

For music generation, I feel a comparison against “StructureNet: INDUCING STRUCTURE IN GENERATED MELODIES” by Medeot et al. (http://ismir2018.ircam.fr/doc/pdfs/126_Paper.pdf) is needed. The method for extracting structure is different, but the goals are very similar. I think it’s important to demonstrate that the method your paper proposes is more effective.

For poetry, I’d like to see a comparison against a method similar to the one described in “Combining Learned Lyrical Structures and Vocabulary for Improved Lyric Generation” by Castro and Attarian (https://arxiv.org/pdf/1811.04651.pdf).

Both of these methods have the advantage of using a relatively simple technique to extract structure from the training data, so I’d like to see evidence that the more complex method proposed in this paper provides an advantage.

My other big concern is the lack of rigour in the evaluation process. This paper both proposes a new method for generating sequences with improved structural characteristics and several new methods for evaluating high level structure. However, very little time is spent validating that these evaluation methods actually accomplish their goals. I would like to see something like a systematic human evaluation of outputs that shows human perception agrees with the automated evaluations.

A few more concerns:

Sections 3 and 4 are very thorough in their descriptions, but I found it difficult to keep all the variables and their uses straight in my head as I was reading through. I think adding some diagrams and a few simplified concrete examples would greatly improve this section.

Section 4.1: Either a citation or more architectural and training details are needed about the LSTM-VAE.

I found Tables 1 and 2 confusing. The bolding doesn’t seem to match up with the best scores. Am I correct that in Table 1, the lowest Low-Level score should be best and the highest High-Level score should be best? For example, shouldn’t the MusicAutoBot row be bolded in the first column? Similar issues for Table 2. I think providing more visual separation between the different scoring mechanisms and explicitly saying “higher is better” or “lower is better” would help clear this up.

Last 2 paragraphs of Section 5: There are a couple references to Figure 5, but there is no Figure 5 in the body of the paper.

The user modifications discussed in the final paragraph of Section 5 are very interesting, and show a real advantage to having interpretable structural constraints. I think this would be much more clear if you included a concrete example of what it looks like to modify the program to produce the new output. Even just a snippet of what the original program looked like, then the modification, then the new output. This is very much related to my comment on Sections 3 and 4: having some concrete examples would really help clarify exactly what’s happening.

**Additional minor feedback**

In the abstract, second sentence, s/generate/generation/.

Section 2, 2nd paragraph, “indicates whether w and w satisfy”. I think the second w was meant to be w’.

Section 4.2, Approach 1. s/rejecting sampling/rejection sampling/.

**Questions for the rebuttal period**

Please address the concerns above.

I’d also like to hear your thoughts on how this can be expanded to music more complex than monophonic folk melodies.

**Recommendation**

My recommendation is to reject due to the concerns listed above, in particular the lack of comparisons to existing structural constraint methods and rigour of the evaluations.

---

### Official Review · AnonReviewer1 · 2020-10-25
**Interesting paper on neural-symbolic models for sequence generation**

**Rating:** 7
**Confidence:** 4

**Review:**

This paper proposes a neural-symbolic deep generative model to generate music and poems. Compared to other natural language sentence generation tasks, songs and poems usually require some structural constraints, e.g., rhythm and rhymes. Therefore, their approach first generates a program represented as a set of relational constraints, then generates sequences that satisfy the constraints. During the training time, given a training example, i.e., a sequence, they formulate the program synthesis task as a combinatorial optimization problem, and thus they can find the required constraints using the Z3 solver. They use the SOTA generative models for both of the two sequence generation domains. They propose several variants of the generation approach, to ensure that the generated sequences satisfy the constraints. They evaluate their approach in terms of both the low-level and high-level structures. To measure the low-level structure, they compute the negative log likelihood of the SOTA generative models. To measure the high-level structure, they apply the Z3 solver to solve the relational constraints of each generated sequence, and train some discriminators to distinguish the generated sequences from the human-written ones in the dataset. For the poetry domain, they also evaluate the diversity of the generated sequences. Generally, the neural-symbolic approach generates sequences with better high-level structures, and the generated samples are more diverse.

I think this paper studies an interesting topic. The high-level framework design is kind of extended from Young et al.'s work on neural-symbolic generative models. However, as discussed in the paper, Young et al. focus on generating simple 2D images with repetitive patterns, and I think it is non-trivial to extend that work to sequence generation domains in this paper. Adding a program synthesis component is a good way of incorporating the structured constraints for generative models, and it is good to see that the approach indeed generates more natural songs and poems than using the pure neural generative models alone.

I have a couple of questions listed below:

1. How scalable is the proposed approach? For example, before the model generates a sequence that satisfies the constraints, typically how many tokens need to be generated during the sampling process, and how long does it take? I see that the authors mention the number of lines for the poetry domain and the number of beats for the music domain, but I would like to know more details about the token-level lengths. Also, how long does it take for the Z3 solver to find the constraints given a sequence?

2. What are the entropy results for the music domain? Also, in Table 1, for the MusicAutoBot results, did you also present the negative log likelihood? If so, the lower should also be better, right? Why is the highest number highlighted?

3. Have the authors conducted any user study? For example, the authors can present both the generated and the human-written songs and poems to people, and see whether they can identify the ones generated by neural models. I don't expect the authors to finish such a user study soon if they haven't done it, but would like to see a discussion if they already have some conclusions.

Writing comments: On page 8, "Figure 5" should be "Figure 2".

---

### Official Review · AnonReviewer4 · 2020-10-26
**General method with interesting ideas, but weak baselines and unconvincing evaluation method**

**Rating:** 6
**Confidence:** 4

**Review:**

### Summary

This paper proposes an approach for generating sequences that possess high-level structure, and in particular structure that can be expressed using hand-crafted symbolic relations. Given a domain and a set of possible relations to consider, this approach first extracts a sequence representation of the relational constraints for each example, then trains a two-stage generative model that first generates relational constraints and then generates the final output conditioned on the constraints. The authors apply their approach to the generation of music and poetry, and show that, compared to simple baselines, the generated outputs are more consistent with human-generated examples (according to other learned models of high- and low-level structure).

The approach is quite general, and seems like an interesting way of imposing structure on sequences. However, I'm not convinced that their approach "significantly improves over state-of-the-art approaches" as the authors claim. The baselines are somewhat limited, and the evaluation metrics that the authors use are difficult to interpret. The authors also do not compare against existing approaches for constrained music and poetry generation, instead comparing only to neural sequence models. Additionally, the generated results still seem to lack global coherence despite having global relational structure.

### Detailed comments

The authors present their approach as an extension of "neurosymbolic generative modeling", which uses program synthesis to extract structure and then trains one model to generate programs and another model to use the output of that program. In this work, the authors consider a different representation of structure, which associates each element of a sequence with a prototype and a set of relations that determine how the element relates to the prototype. The prototypes in this approach are subcomponents of the full output (for music, they are measures, and for poetry, they are lines), and the relations are hand-engineered (rhyme scheme and meter for poetry, and various music theory concepts for music). This relational graph is still referred to as a "program", and the extraction of the graph as "program synthesis", but I'm not sure that's the appropriate terminology; the graph isn't executable, it is just a symbolic object.

The generative process that the authors propose is to first generate constraints (expressed as a sequence), then generate prototypes, and finally generate output conditioned on the constraints and prototypes. Each of these components is trained separately. Given a dataset, a maximal set of constraints is extracted from each example by using an SMT solver. Afterward, a VAE is trained to produce constraints, and a pretrained model is used to generate prototypes. Finally, examples are sampled conditioned on these constraints, either by doing rejection sampling on a pretrained model or by training a new generative model to generate outputs conditioned on constraints (or, in some cases, by doing combinatorial search with heuristics).

The authors cite a large number of works in the realm of neurosymbolic machine learning, but are not as thorough regarding music and poetry generation, both of which are subfields of their own right. For music generation, there are existing approaches for trying to learn long-term structure, such as Music Transformer [1], hierarchical VAEs [2], and StructureNet [3]. For poetry, structure can be learned using methods like weighted transducers [4], and there are also previous works on using explicit constraints to generate poems [5]. The authors do not compare with any of these methods, and the ones they do compare against aren't necessarily representative of "state of the art".

I'm also not convinced that the metrics the authors use are adequate for judging performance. For "low level" structure, the authors report the (negative) log-likelihood scores given by a pretrained sequence model, with the assumption that higher log-likelihood means better low-level structure. But it has been shown that log-likelihood of models does NOT directly correspond to what humans find "good" (see for instance [6]), and models will in some cases assign higher log-likelihood to data far outside of their training set (e.g. [7]), which means that higher log-likelihood isn't necessarily a good proxy for human-like structure. It would be more standard to report log-likelihood of the human test data under the proposed model, instead of log-likelihood of the proposed model's outputs under some pre-trained model.

For high-level structure, the authors extract relations using their method, then train a separate classifier to discriminate between human and generated examples based on those extracted relations as features. But this seems to be slightly unfair, since their model is trained directly to maximize this objective, whereas other models do not use this at all. The presentation of these results is also quite confusing (see below).

In a more qualitative sense, while it is true that the generated poems do have reasonable rhyme and meter, the semantic content of the poems is disjointed and does not have a sense of global coherence. I wonder if the proposed approach is satisfying the hand-engineered relations at the expense of losing the semantic structure that is harder to measure.

### Questions and suggestions

Abstract: Should "generate" be "generation"? Also, the sentence "To train model (i), ... resulting relational constraints" is a bit unclear. Is that whole sentence about model (i) or is the last part referring to model (ii)?

In figure 1, what do the colors of the lines represent? It seems that there are both green and purple relational constraints, but these colors aren't explained.

The notation in sections 2 and 3 is a bit unclear. The set $\mathcal{C}$ is used before being defined, and it isn't clear what the difference is between $c$, $f$, and $\Phi$; are those all just representations of the same thing? The sum in the equation at the bottom of page 3 is over $r \in \mathcal{R}$ but $r$ doesn't appear in the equation, only $c$ does.

It seems somewhat restrictive to assume that each line/measure has only one prototype, since it's possible for line C to share rhyme with line A but share meter with line B. Have you considered a more general formulation of constraints?

I was surprised that the prototype subcomponents are represented as part of the constraints but are generated independently from the set of relations. It seems like those prototype subcomponents could just as easily be left out of the constraint sets and instead just be generated as part of the second step. Is there a reason that they need to be part of the constraints?

The optimization problem in 3.2 seems underdetermined. In particular, if X and Y have some relation between them, it doesn't seem like there's anything to determine which of the two should become the prototype. Is one of those simply chosen arbitrarily?

In section 4.1, $p_\phi(c|z)$ is referred to as "a VAE", which seems slightly inaccurate; technically the VAE includes the prior, encoder, and decoder, not just the decoder.

In section 5, the description of how A3 is applied to the music domain seems to be missing from the first paragraph.

The results in table 1 are very difficult to understand, especially for the high-level classification task. Based on the appendix, it seems that "RF Disc." is measuring classification accuracy, for which lower values are better (because more humanlike outputs are harder to classify). But "CGN Disc." is measuring cross-entropy loss, for which higher values are better. This was very difficult to follow. I think it would be clearer to just report accuracy for both, or, at the very least, describe what each of the columns are measuring. I also don't understand the Human row of this table. Is this comparing one set of human examples to a different set of human examples? For RF, 50% seems like the expected chance accuracy, but distinguishing humans from humans is apparently possible 51% of the time? For GCN, the value 0.69 makes me think this must be measured with a natural logarithm, is that correct?

The statement "our approach outperforms BERT as a language model in terms of its own score" seems unfair due to the reasons described in the previous section. We would not expect a language model to assign the highest likelihood to all of its own predictions, since it is supposed to describe a full distribution of outputs. The likelihood assigned by a language model isn't a measure of goodness, just a measure of how predictable that sequence is by the model. At most, perhaps this indicates that the proposed approach is more predictable than real-world natural language, so BERT is able to predict it better. (Also, is the BERT used to score the models the same as the BERT used to generate? Or is one of them fine-tuned and the other left as-is?)

### References

[1] Huang, Cheng-Zhi Anna, et al. "Music transformer: Generating music with long-term structure." International Conference on Learning Representations. 2018.

[2] Roberts, Adam, Jesse Engel, and Douglas Eck. "Hierarchical variational autoencoders for music." NIPS Workshop on Machine Learning for Creativity and Design. Vol. 3. 2017.

[3] Medeot, Gabriele, et al. "StructureNet: Inducing Structure in Generated Melodies." ISMIR. 2018.

[4] Hopkins, Jack, and Douwe Kiela. "Automatically generating rhythmic verse with neural networks." Proceedings of the 55th Annual Meeting of the Association for Computational Linguistics (Volume 1: Long Papers). 2017.

[5] Toivanen, Jukka, Matti Järvisalo, and Hannu Toivonen. "Harnessing constraint programming for poetry composition." The Fourth International Conference on Computational Creativity. The University of Sydney, 2013.

[6] Meister, Clara, Tim Vieira, and Ryan Cotterell. "If beam search is the answer, what was the question?." arXiv preprint arXiv:2010.02650 (2020).

[7] Nalisnick, Eric, et al. "Do deep generative models know what they don't know?." arXiv preprint arXiv:1810.09136 (2018).


---

## Post-revision update

I cannot seem to add a new comment at this time, so I am editing this review instead. The updated paper seems to be an improvement, although some of my original concerns remain.

Improvements:

- Thank you for adding the additional baselines, which do give more context as to how this approach relates to prior work.
- The section describing $c$, $f$, and $\Phi$ is clearer in the revision, and is much easier to follow.
- I also appreciate the clarification about nondeterministic choices in Z3.
- The overly-strong claims about outperforming state-of-the-art approaches have been qualified appropriately.
- Some of the confusing details regarding the evaluation method have been moved from the appendix to the main text, which makes them much easier to find.

Remaining high-level concern:

- I'm still not convinced that the results of this evaluation method are that meaningful.
  + NLL under a pretrained model doesn't necessarily imply better low-level structure, for reasons stated in my initial review.
  + Ease-of-discrimination of your extracted constraints doesn't necessarily imply better high-level structure, especially since the proposed model is trained on those constraints directly.

Other remaining issues in the revision:

- A few comments from my initial review have not been addressed:
  + "To train model (i), ... resulting relational constraints" remains unclear
  + In section 4.1, $p_\phi(c|z)$ is referred to as "a VAE" but is just a small part of a VAE
  + In section 5, the description of how A3 is applied to the music domain is missing
  + Table 1 is still difficult to interpret, in particular regarding the higher-is-better vs lower-is-better columns, and the (new in this revision) bolding of the human data, which I don't quite follow.
- Figure 1's caption now states that rhyme and meter constraints are the reason for the green and purple edges, but this is confusing because there are purple edges between poem lines that don't have meter constraints.
- Regarding semantic content, the authors state in their response that "The semantic content of the poems (or lack thereof) reflects defects in the underlying deep generative models". But Section 4.2 approach 1 seems to imply that each of the lines are sampled independently of one another, which seems like a strong limitation that the underlying models do not have. Perhaps the notation is unclear, and the model does get the full input context; if this is the case, I would suggest revising the notation.

I have raised my score from 5 to 6 in light of the improvements, and with the understanding that my concerns in "Other remaining issues in the revision" could be fixed in a final version of the paper. I'm still borderline on accepting the paper, however, due to my concern (shared with Reviewer 3) about how meaningful the evaluation results are and whether they match what humans mean by high-level and low-level structure.

---

### Official Review · AnonReviewer2 · 2020-10-30

**Rating:** 6
**Confidence:** 4

**Review:**

# Overall summary
This paper proposes a generative model for sequential data structured which uses latent relational constraints and conditions upon them to generate the actual sequence.  The authors propose that such a model can be better suited for data like poetry or music where such constraints are an important feature of the data.

# Strengths
- The method provides a strong and useful inductive bias for modeling sequences which we can expect to have strong relational constraints, without having to learn them from scratch.
- The method incorporates a discrete optimization-based component which can lead to much more interpretable results.

# Weaknesses
- The method requires significant hand design of the constraints in order to work well. This may be difficult or counter-effective when the constraints are not easily described for the domain at hand.
- It is not entirely clear when the relational constraint optimization will produce good results. There is no theory presented about what it might find.
- The empirical results presented are not especially strong.

# Comments on the evaluation
It should be possible to estimate a lower bound on the log likelihood of the proposed model on a held-out test set, rather than using different models. After all, the goal is that the proposed method can model these sequential data with relational constraints better than the existing methods. Therefore, it's not clear that MusicAutoBot, MusicVAE, BERT, etc. putting higher or lower likelihood on samples from the proposed method should necessarily be a strong indication of the method's quality.
To provide a tighter lower bound, it's possible to sample z multiple times and average, as in IWAE: https://arxiv.org/abs/1509.00519
Furthermore, if the priors about the data that the proposed method presupposes are correct, then it should be possible to show greater data efficiency than the baseline methods.
I would be willing to revise the score if the authors can provide further explanation/data about this.

---

### Author Response · Authors · 2020-11-25
**Response to Reviews**

Discussion of new results:
Based on the feedback from the reviewers, we significantly updated our paper and expanded our experiments. The changes include the following:
- Improved writing. We have done our best to improve the clarity of the paper and incorporate responses to reviewer questions. We have added a detailed example at the end of Section 3.1, together with remarks discussing our design choices. We have also added a discussion of the optimization problem at the end of Section 3.2.
- Adding baselines that leverage structure, including two for music (StructureNet and Motifate), and one for poetry (RichLyrics).
   - We have added a comparison to StructureNet; we find it is comparable to GraphVAE on non-structure metrics (i.e., NLL according to a probabilistic model), but substantially worse on structure metrics (i.e., RF or graph discriminator). Motifate is much worse in terms of structure metrics, though we could not evaluate it in terms of NLL since it is specialized to 3-beat measures (whereas our dataset only contains songs with 4-beat measures). RichLyrics performed strictly worse than our approach according to all our metrics (except Shannon entropy). We could not find any constraint-guided approaches to generating poetry that fit our task, except for a hybrid neural/constraint-based tool, https://github.com/summerstay/true_poetry, which timed out after 10 hours for a single example.
- Standardizing the metrics for MusicAutoBot, MusicVAE, and our Graph-VAE so that all 3 are negative log-likelihood.
   - Before, we were normalizing the metric reported by MusicAutobot by length, so it was much smaller. Now, for all models, we are reporting the negative log-likelihood (NLL). In particular, our approach produces the lowest NLL on the held-out human dataset. The remaining approaches are not probabilistic, so we cannot obtain an NLL of the human data.
- Comparison in terms of NLL on held-out human dataset..
   - We have reported NLL results on the held-out human dataset for MusicAutobot, MusicVAE, and our GraphVAE. We find that our GraphVAE approach performs best. For the remaining models, we cannot obtain estimates of the NLL. For example, several of the other approaches rely on constrained sampling (including StructureNet, and our approach for the poetry domain); we would have to integrate over all possible constraints to obtain an estimate of the NLL on the held-out human dataset, which is computationally intractable.
- Graph discriminator.
   - We clarify that the Graph discriminator and random forest are measuring cross entropy loss on a balanced dataset of 50% human held-out data and 50% generated data

---

### Decision · Program_Chairs · 2021-01-07
**Final Decision**

**Decision:**

Reject

**Comment:**

This paper was pretty borderline, but ultimately I am recommending rejection, for the following reason:

The two most negative reviewers (in terms of original score) were concerned about both the quality of the evaluation
and whether the evaluation metrics actually meant what the paper claimed they meant.
The authors did make a good faith effort to update the paper to respond to some of the concerns about quality,
but R4 (who has read the rebuttal) is still not convinced that the results of the evaluation are meaningful,
and I think I agree with their concern (and I don't see any attempt to address that concern in the rebuttal?).
I view (maybe naively) the goal of this review process as being mostly a correctness check,
and I don't think this paper has passed the correctness check to my satisfaction or to the satisfaction of the majority of reviewers.

However! This is a fixable issue. The paper is definitely cool and interesting, and I would urge the authors to think harder
about what sort of evaluation makes sense here and resubmit to the next suitable machine learning conference.